# The Development of a Food Frequency Questionnaire for the Assessment of Ultra-Processed Food Consumption in the Italian Adult Population: Protocol for a Validity and Reproducibility Study

**DOI:** 10.3390/nu16223896

**Published:** 2024-11-15

**Authors:** Erica Cardamone, Francesca Iacoponi, Federica Fiori, Michela Marinoni, Umberto Agrimi, Marco Silano, Maria Parpinel

**Affiliations:** 1Department of Medicine-DMED, University of Udine, 33100 Udine, Italy; federica.fiori@uniud.it (F.F.); maria.parpinel@uniud.it (M.P.); 2Unit of Human Nutrition and Health, Department of Food Safety, Nutrition and Veterinary Public Health, Italian National Institute of Health, 00161 Rome, Italy; francesca.iacoponi@iss.it (F.I.); umberto.agrimi@iss.it (U.A.); 3Branch of Medical Statistics, Biometry and Epidemiology “G. A. Maccacaro”, Department of Clinical Sciences and Community Health, University of Milan, 20122 Milan, Italy; michela.marinoni@unimi.it; 4Department of Cardiovascular, Endocrine-Metabolic Diseases and Aging, Italian National Institute of Health, 00161 Rome, Italy; marco.silano@iss.it

**Keywords:** food frequency questionnaire, ultra-processed foods, NOVA classification, validity, reproducibility

## Abstract

Background: Over the last decade, while studies on the detrimental effects of ultra-processed food (UPF) consumption have increased, methodological limitations on the quality of available evidence have emerged. Starting from a critical reassessment of the NOVA classification, this project will aim to develop and validate a food frequency questionnaire (FFQ), which is based on the processing of consumed foods and specifically designed to estimate the UPF consumption and total dietary intake of macro- and micronutrients in the Italian adult population. Methods: This study will take place in selected workplaces and include healthy males and females aged ≥18 years, residing in Italy and with Italian citizenship. The FFQ will be online, voluntary, self-administered, semi-quantitative, and designed to assess food intake over the past year and distinguish between industrial, artisanal, and home-made products. This project will consist of two phases. First, a pilot study will be conducted to obtain the final version of the FFQ. The current food consumption of the target population will be investigated, through a 24 h dietary recall, and the face validity of the new tool will be tested. The second phase will involve at least 436 participants. To assess reproducibility, the FFQ will be administered twice (at an interval of 3–10 months), and the test–retest method will be used. A 7-day weighed dietary record (WDR) will also be completed after each FFQ administration. To evaluate criterion validity, data from the two WDRs will be compared against those from the first FFQ administration. Conclusions: The results will provide a new valid tool focused on food processing, potentially useful for future studies.

## 1. Introduction

Over the past few decades, Mediterranean countries have experienced a progressive shift away from traditional healthy dietary patterns, such as the Mediterranean Diet (MD) [1,2,3]. Globalization, population growth, urbanization, and lifestyle changes are just some of the factors that have contributed to this phenomenon [4,5]. A dietary transition occurred with the rapid diffusion of the so-called ‘Western Diet’, a dietary pattern mainly characterized by a high consumption of energy-dense foods that are High in Fat, Sugar, and Salt (HFSS) and relatively cheap and highly palatable ready-to-eat products [6]. The shift towards Westernized dietary patterns, together with a sedentary lifestyle, is one of the main determinants of increasingly evident conditions, such as overweight, obesity, and Noncommunicable Chronic Diseases (NCDs) [7]. This critical scenario also characterizes Italy, as confirmed by data from national surveillance systems of all age groups in the Italian population [8].

In this context, a new way of thinking about food classification has emerged, no longer based on nutrient content but on food processing [9]. Although there is no universally agreed definition of processed foods, several classification systems based on the level of food processing have been developed worldwide [10,11]. Sadler and colleagues [11] in a critical analysis of existing food classification systems identified four different dimensions used in the conceptualization of processed foods: (1) the extent of change from the natural state, (2) nature of change, (3) place of change, and (4) purpose of change. In 2023, the Scientific Advisory Committee on Nutrition (SACN) published a statement [10] in which it was evaluated how the four dimensions identified by Sadler et al. [11] had been taken into account in the development of existing classification systems. The only evaluated classification developed considering all four dimensions of processing is the NOVA classification, developed by Monteiro et al. in 2009 [12], which represents the best-known and most widely used system in the scientific community. The NOVA classification divides foods into four categories: (I) unprocessed or minimally processed foods, (II) processed culinary ingredients, (III) processed foods, and (IV) ultra-processed foods (UPFs) [13,14].

The latter is a heterogeneous group of foods, defined as formulations of ingredients, mostly of exclusive industrial use, made by a series of industrial processes, many requiring sophisticated equipment and technology. The processes and ingredients used for their production are designed to create hyper-palatable, low-cost, and long-life ready-to-eat products [13,14]. Increasing evidence focuses on UPF consumption and its potential effects on human health, suggesting an association between a high UPF intake and the development of NCDs and overweight and obesity [15,16]. Furthermore, several limitations have been identified among the currently available studies, recognizing that further research is needed to better understand this relationship [10]. These limitations include, for example, the food classification system used, the almost exclusively observational nature of these studies, the different methods and cut-offs of reporting UPF intake across these studies, and the fact that it is unclear whether the adverse health associations are due to or independent of the nutrient content of many UPFs [10]. Another great methodological limitation of this evidence is represented by the tools used to assess UPF consumption, which were not specifically developed for this objective [10,17]. To date, few studies have been conducted with the aim of validating tools for the assessment of UPF consumption. Among these, there was only one study conducted in Italy, which led to the development of a food frequency questionnaire (FFQ) based on the NOVA classification [18]. To the best of our knowledge, there are no data showing that the tools currently available in Italy are validated for estimating the total daily intake of nutrients, particularly micronutrients, and allow for a precise identification of processed foods and UPFs consumed (e.g., through questions about the brand of products consumed).

Controversial opinions and conceptual limitations have also been raised about the NOVA classification, such that several national and international institutions have called on stakeholders not to use or critically reconsider the current classification and terminology [10,17].

In view of this, the objective of the present manuscript is to describe the protocol of a study that will aim to develop and validate an FFQ, which is based on food processing and specifically designed to estimate the UPF consumption and daily intake of macro- and micronutrients from the total diet, in the Italian adult population, starting from a critical reassessment of the NOVA classification and referring to previously validated tools.

## 2. Materials and Methods

### 2.1. Ethics and Study Design

This project was approved by Ethics Committee of the Italian National Institute of Health (approval n AOO 0040859 on 26 September 2024). This will be a multicenter study, promoted and conducted by the Italian National Institute of Health and the University of Udine, consisting of two consecutive phases:A pilot study to evaluate the current food consumption of the target audience and test the face validity of the new FFQ;A validation study to test the criterion validity and reproducibility of the new tool.

### 2.2. Population

This study will be conducted in selected workplaces (e.g., university, research institute) and will include the employees and students of the institutions involved. Participants will be healthy males and females aged ≥18 years, residing in Italy and with Italian citizenship. Individuals with at least one concomitant pathology (e.g., hypertension, diabetes, cardiovascular diseases, cancer), following one or more restrictive diets (e.g., gluten-free diet, hypocaloric diet), in specific physiological states (e.g., pregnancy, breastfeeding), not residing in Italy, and without Italian citizenship will be excluded. Participation, subject to informed consent, will be free and voluntary. The invitation letter will be disseminated via the mailing lists of the selected organizations. Participants will be invited to involve family members and acquaintances, aged ≥18 years, if interested and willing to take part in this project.

The initial phase of this study will involve the recruitment of a convenience sample of 20–50 participants from selected workplaces, which will not be included in phase 2.

In the second phase, a minimum of 436 individuals will be recruited. There is evidence that a sample size of 100–200 is reasonable for validation studies and that studies including 200–300 participants can provide informative data on questionnaire validity [19]. Based on this, the sample size was calculated by considering the drop-out rates of similar studies conducted previously (34%) [18,20] and the prevalence of overweight and obesity in the Italian adult population (44%) [21], in order to have a number of obese and overweight subjects to represent the distribution in the Italian population.

### 2.3. Definition of Modified NOVA Classification

The NOVA classification was chosen as the basis for this work as, of all existing systems, it is the most widely used in the literature and takes into account all four core themes of food processing mentioned above. At an early stage, a critical reassessment of the NOVA classification was carried out, which was necessary during the development of the questionnaire and subsequent analyses.

An in-depth study of the NOVA criteria and the limitations of this classification was conducted. It was observed that among the main criticalities of this classification is the fact that it is a qualitative system, which does not properly consider the level and degree of food processing [17]. The NOVA criteria do not make it an easy system to interpret and apply and do not allow for robust and functional food assignments [22]. In addition, the NOVA categories are broad, especially the UPF group, and capture a wide range of foods and beverages [10]. Based on these considerations, it was decided to propose a modification of the NOVA classification that would allow for a more precise categorization of food and beverages.

Recent studies suggest that a new classification system combining the level of processing and nutritional profile may be a valid alternative to existing methods [23,24,25]. In accordance with this evidence and other previously developed systems available in the literature [26], a modified version of the NOVA classification, called ‘modified NOVA’ (mNOVA), was developed. In the mNOVA, the groups 3 (processed foods) and 4 (UPFs) are in turn divided into two subgroups of foods (3a and 3b, 4a and 4b), based on the salt, sugar, and fat thresholds recommended by the Food Standard Agency (FSA) [27] (Figure 1).

### 2.4. Questionnaire Development

The questionnaire, called ‘mNOVA FFQ’, will be self-administered and completed on the web application ‘Walnut-BDA’, a tool for building, administering, and analyzing food frequency questionnaires and dietary records, which was developed by the University of Udine.

Before accessing the questionnaire, in accordance with the European Commission General Data Protection Regulation (679/2016), consent to collect and process personal data will be given electronically (ticking the corresponding box) on the web-based platform.

The system will associate each respondent with an automatically generated code. At the end of the questionnaire, the code will be sent by e-mail, with a request to keep it for future use, together with the notification of completion. In order to guarantee the pairing of data obtained from multiple administrations in the various phases of this study, the e-mail addresses of respondents, associated with the generated unique codes, will only be stored until the end of the data collection.

The questionnaire will include 3 sections: the first will consist of questions on socio-demographic characteristics, the second on anamnestic information, and the last on dietary data. In detail, the first and second sections will include 9 main questions each on demographic information (e.g., age, sex, citizenship), socio-economic status (e.g., qualification, employment status, family income), lifestyle (e.g., smoking habits, physical activity, adherence to special diets, health status), and the latest measurements of weight, height, and blood pressure taken by a general practitioner or other health professional and fasting blood glucose (measurement within the last 6 months). These will be useful for confirming the eligibility of participants, for stratifying the sample, and for potential future perspectives. The third and final section will include questions from the new mNOVA FFQ.

An initial draft of the mNOVA FFQ was developed from tools previously validated in the Italian population and available in the literature (Figure 2) [18,20,28].

This is a semi-quantitative FFQ, based on the level of food processing, designed to assess the food intake over the past 12 months and to distinguish between industrial, artisanal, and home-made products. The draft is organized into 8 parts, corresponding to 8 food and beverage groups: (A) beverages; (B) milk and dairy products; (C) cereals; (D) meat, fish, and eggs; (E) vegetables, legumes, and fruit; (F) oils, fats, and seasonings; (G) sweets; and (H) other. To date, it includes 52 main yes/no questions on specific foods and beverages commonly consumed in Italy. In the case of a yes answer, additional questions on portion size (choosing between small, medium, and large portions, using illustrative pictures and household units, according to national quantitative standards for portions), frequency of consumption (choosing from 10 different options, ranging from ‘never’ to ‘>5 times per day’), and type of products usually consumed (e.g., brand, cooking method, specific traditional Italian recipes) will be administered. Overall, the FFQ collects information on the usual consumption of 108 total food items. As shown in Figure 2, the final version of the mNOVA FFQ will be developed after the pilot study, reviewing the tool through dietary intake data from the Italian national dietary survey on adult population (IV SCAI ADULT) [29] and data collected with the 24 h dietary recall (24HR) in a target audience during the first phase of this project.

### 2.5. Phase 1: Pilot Study

The pilot study will be conducted on a convenience sample of potential participants in the second phase. Data will be collected online, on the above-mentioned web application (Walnut-BDA), through the self-administration of the new FFQ draft and the administration by a trained interviewer of a 24HR (Figure 3).

Selected participants will be contacted by e-mail with a request to complete the questionnaire, after the acceptance of informed consent. The administration of the mNOVA FFQ draft will be necessary to test the face validity, i.e., the extent to which a test appears to measure what is intended to measure, in order to improve the acceptability, relevance, and quality of the new tool [30].

After the completion of the questionnaire, a trained interviewer will conduct an online interview in order to capture each respondent’s understanding and interpretability of the mNOVA FFQ (e.g., initial instructions, questions, options for frequency and portion size of consumption, descriptions of food items) and to administer a 24HR. The recalls will be analyzed to provide a list of the food most consumed by the respondents in the 24 h prior to the interview so as to better understand the eating habits of the population targeted by the FFQ and to possibly adjust the closed list of food items.

The data collected in this first phase of the project will be analyzed and used to define the final version of the questionnaire to be validated in the next phase.

### 2.6. Phase 2: Validity and Reproducibility

Once the mNOVA FFQ is defined, the validity and reproducibility study will start and will involve the estimated sample of at least 436 subjects.

In order to assess the reliability and reproducibility of the questionnaire, it will be administered online on two occasions, in two different seasons, and the data obtained will be compared (Figure 4).

Three to ten months after the first administration (T0), participants will be contacted again, using the e-mail addresses provided, and, on a voluntary basis, may proceed to the second administration (T1). In the week subsequent to the administration of each FFQ, participants will be asked to fill in a 7-day weighed dietary record (WDR) on the Walnut-BDA web application. This will be used to compare the data collected with the FFQ and those collected with a reference method, such as the WDR, to test the criterion validity of the mNOVA FFQ, i.e., its ability to measure the outcome for which it was designed [31].

### 2.7. Statistical Analysis

Each participant will be identified by the random unique code generated by Walnut-BDA.

This web application will automatically decode and analyze the food data collected with the tools, estimating the daily intake of total energy and nutrients, using the Food Composition Database for Epidemiological Studies in Italy (BDA) [32].

Analyses will be based on the daily food intake for each category of the mNOVA classification (g/day or % total energy) and the daily intake of total energy (kcal/day), macronutrients (g/day and % total energy), and micronutrients (g/day or mg/day).

After checking for missing values, inconsistencies, and outliers, descriptive analyses will be performed (mean and standard deviation, median and interquartile range, absolute frequencies and percentages, depending on variable distribution).

In order to assess the reproducibility of the mNOVA FFQ, the test–retest method will be used. Specifically, the Pearson Correlation Coefficient (or the Spearman Correlation Coefficient, depending on the data distribution) will be calculated to evaluate the association of the data collected through the questionnaire at T0 (baseline) and T1 (after 3–10 months). The intraclass correlation coefficient (ICC) and 95% confidence interval (95% CI), using the two-way mixed effects model, will also be calculated.

Criterion validity will be assessed by comparing data obtained in the first administration of the mNOVA FFQ (T0) and an average of the data obtained in the two WDRs. The Pearson Correlation Coefficient (or Spearman Coefficient), ICC, and 95% CI (using mixed effects models) will also be calculated. In order to further explore the validity of the questionnaire, Bland–Altman plots will be created to produce quantitative estimates of agreement between the two methods.

A *p*-value < 0.05 (two-tailed) will be considered statistically significant. StataSE V.18 for Windows (StataCorp) and R Core Team (2024) software 4.4.1 will be used for all statistical analyses.

## 3. Discussion and Future Perspectives

In the context where growing evidence shows the detrimental effects of the high consumption of UPFs [25] and associates it with the increased occurrence of obesity, overweight, and NCDs, methodological limitations and uncertainties about the quality of the available studies remain [10]. This highlights the need for further research, specifically to develop a valid, reproducible, population-specific, and up-to-date tool for the evaluation of UPF consumption.

The project described in this manuscript will focus on developing and validating an FFQ, specifically tailored to assess the UPF consumption and simultaneously estimate the total dietary intake of macro- and micronutrients in the Italian adult population.

Most of the existing tools for estimating UPF consumption are based on the NOVA classification [18,33,34,35,36,37,38]. There was a need to develop an FFQ based on a classification system that would attempt to achieve a more accurate categorization of food and beverages. The critical analysis of the NOVA classification led to the development of the mNOVA classification, which combines the level of food processing with its nutritional profile (see Figure 1). This was proposed in order to especially overcome the limitation of the NOVA classification related to the wide variety of food products included in the broad NOVA categories, in particular in groups 3 (processed foods) and 4 (UPFs). Confirming that UPFs are not a homogeneous entity in terms of their nutritional quality and health impact, there are recent data observed in large prospective cohorts in the US and Europe [16,39,40,41]. Analyses were conducted on the consumption of various subgroups of UPFs in association with the risk of multimorbidity, showing a direct association with UPFs, such as artificially and sugar-sweetened beverages and animal-based products, and an inverse or null association with foods such as cereals and plant-based products. In this respect, the mNOVA classification, based on the FSA thresholds, could enable a better differentiation of the various types of processed foods and UPFs.

Based on mNOVA classification system and tools previously validated in Italy, a draft of the self-administered online questionnaire (mNOVA FFQ) was designed to assess the food intake over the past year and to distinguish between industrial, artisanal, and home-made products.

This project will consist of two consecutive phases. Initially, a pilot study will be conducted with the aim of obtaining the final version of the questionnaire to be validated in the next phase. The second phase will focus on testing the validity and reproducibility of the mNOVA FFQ.

Compared to the instruments currently available in Italy, the mNOVA FFQ consists of only 52 main yes/no questions to make the compilation easier and smoother. Indeed, it is important that the length of an FFQ be defined in order to ensure an appropriate balance between accuracy and time of use [28]. Additional questions on portion size, frequency of consumption, and type of products usually consumed (e.g., brand, cooking method, recipe) will be asked only if the initial questions are answered in the affirmative. This may allow for a clear identification of the food and consequently a reliable estimate of macro- and micronutrient intake. Furthermore, questions on the brand name of the products consumed will make it possible to trace the nutritional label and thus classify foods and beverages accurately according to the mNOVA groups.

The strengths of this study include its inclusion of an accurate and rigorous validity and reproducibility methodology and a large estimated sample size. Furthermore, the development and analysis of the questionnaire will be based on an updated classification system (mNOVA) that takes into account both the level of food processing and nutritional content. Other strengths are the rapidity, simplicity, and affordability of the web-based collection of data.

A selection bias could be detected, since the study participants will be volunteers and recruited in selected workplaces. In order to minimize this bias, it was decided to extend the involvement to family members and acquaintances outside the expected setting. Moreover, as this project is relatively long and demanding, there is a risk of a high drop-out rate at follow-up. To mitigate this, the sample size was estimated by considering the drop-out rates of studies previously conducted in Italy. In conclusion, the self-administration of the questionnaire could lead to recall and misclassification bias. However, the mNOVA FFQ will be validated against a WDR, which is considered the gold-standard method for dietary assessment [42]. In addition, check questions were added in the questionnaire to verify the accuracy of the data [43].

Looking forward, at the end of the validation process of the new tool, an analysis of the data collected during the second phase of this project will be taken into consideration, with the aim of analyzing the health impact of UPF consumption. In particular, the association between high UPF consumption and certain health outcomes (e.g., body mass index, blood pressure, fasting glycemia) will be studied in the subgroup of working participants. This analysis comes from the hypothesis that there is a different prevalence of UPF consumption among workers compared to non-workers and the intention to test whether there is a difference in association between UPF consumption and health outcomes between the two groups. This is supported by data from a recent cross-sectional survey on adherence to the MD, conducted by the Italian National Institute of Health [3], which showed a lower adherence to the MD among working respondents, suggesting a higher intake of ready-to-eat products, and of UPFs in general, in this specific population group.

## 4. Conclusions

The results of the present project will provide a valid and reproducible new tool, focused on food processing and potentially useful for future studies. To this end, the findings will be disseminated and published in a scientific journal to make this study reproducible in other settings and validate the instrument in other countries. In the near future, such an epidemiological tool may be a practical support to better understand the UPF category and its impact on human health in the Italian adult population.

## Figures and Tables

**Figure 1 nutrients-16-03896-f001:**
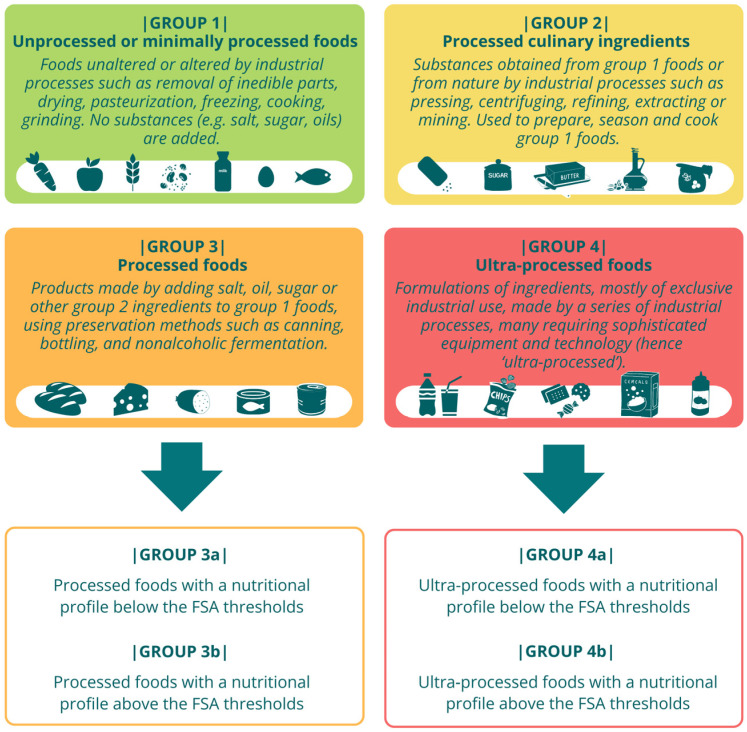
mNOVA classification system. FSA, Food Standard Agency.

**Figure 2 nutrients-16-03896-f002:**
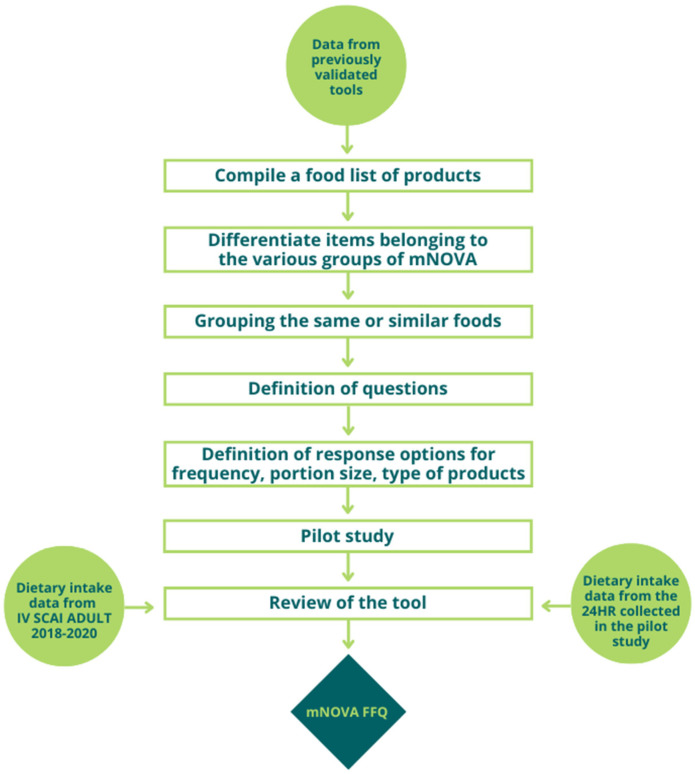
A flowchart of the development of the mNOVA FFQ. 24HR, 24 h dietary recall.

**Figure 3 nutrients-16-03896-f003:**
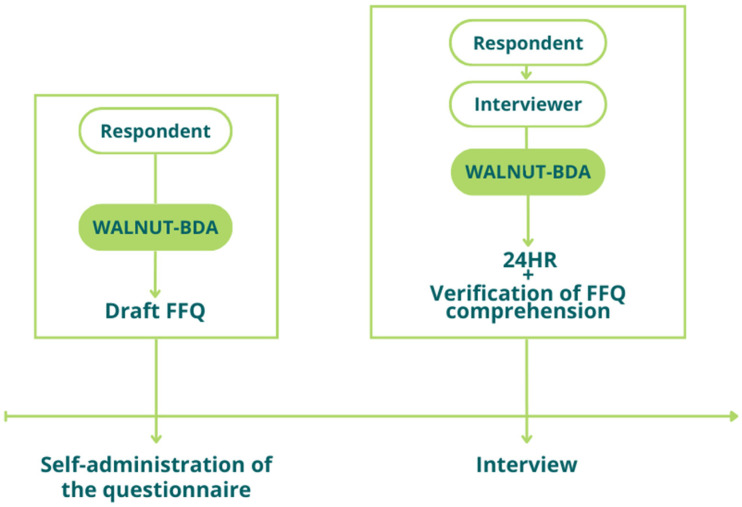
Design of phase 1. FFQ, food frequency questionnaire; 24HR, 24 h dietary recall.

**Figure 4 nutrients-16-03896-f004:**
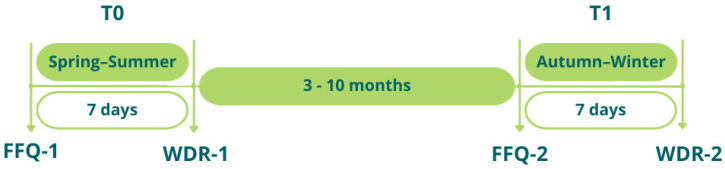
Design of phase 2. FFQ, food frequency questionnaire; WDR, weighed dietary record. Adapted from Decarli A. et al. Ann Epidemiol. 1996; 6(2):110-8 [20].

## Data Availability

The original contributions presented in the study are included in this article; further inquiries can be directed to the corresponding author.

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
