# Peer review of "The Development of a Food Frequency Questionnaire for the Assessment of Ultra-Processed Food Consumption in the Italian Adult Population: Protocol for a Validity and Reproducibility Study"

_nutrients, 2024, doi:10.3390/nu16223896_

Round 1

Reviewer 1 Report

Comments and Suggestions for Authors

Dear Authors,

The manuscript submitted for review (nutrients-3304582) is interesting. However, if I understand correctly, these are only assumptions for validation, not the validation itself. That is why I wonder if the title of the manuscript is appropriate. This, of course, changes its usefulness for other scientists, as I don't know if it's a good research tool.

The beginning of the discussion is a repetition of information from the Introduction and the Methodology.

How will the validation group of respondents be selected? The description shows that it will be a random group obtained on the Internet. It is unknown whether it will be possible to get a diverse group in terms of gender, age, or financial status, which, in the case of ultra-processed food, is of great importance for the results, especially since the research group is only 436 people.

Overall, the study is interesting, and I would be happy to conduct it myself, but more information about it is needed.

Reviewer

Author Response

Dear Reviewer, 

Thank you for your comments. We hope we have understood and answered all comments fully. 

Kind regards

Reviewer 2 Report

Comments and Suggestions for Authors

The manuscript describes the planned development of a food frequency questionnaire (FFQ) calibrated for the Italian Population. The authors have detailed the planned development process. However, the manuscript would benefit from a strengthened theoretical background and argumentation of why the existing validated instruments for the Italian population as mentioned in the methods section are not sufficient – i.e. why a new instrument is needed. The introduction mentions several limitations of the existing tools for assessing the consumption of UPF (ultra-processed foods) and their potential association to NCD. It is unclear how the new proposed FFQ questionnaire aims to overcome these limitations – this should be described clearly in the manuscript. Also, the authors should consider strengthening the argument of why – despite criticisms of the NOVA framework – this was used as the basis of the work. What does the abbreviation of NOVA stand for? 

The authors also mention that there is no universally agreed definition of processed food. Authors should consider providing the readers with context of how the different definitions compare – this would strengthen the argument of selecting the NOVA framework despite the stated criticism. 

The authors also mention that they have critically reassessed the NOVA framework. What this meant in practice and how this was done should be clearly described. The reassessment led to the NOVA framework. 

Despite the manuscript presenting a study protocol, clear aims should be included. 

Inclusion and exclusion criteria – The study population and rationale for the inclusion and exclusion criteria should be better explained. If especially universities or research institutes are targeted, the population may not be representative of the Italian population (e.g. gender, age, socioeconomic background, etc.). The authors also specifically mention the Italian nationality as an inclusion criterion, this should be elaborated – i.e. can the FFQ be reliably applied among populations with known residency and nationality? Furthermore, it is unclear what is meant by concomitant pathology. Were individuals with chronic (e.g. high BP) ineligible for taking part (line 98)? 

The content of the three questionnaire parts should be described with greater detail (line 142). Also, it appears that participants are asked to report information e.g. fasting blood glucose, that may not be done routinely (line 149). It is also unclear what the purpose of this information is for the questionnaire development.

Line 104 – This sentence is unclear. Could the authors please check this? 

A large part of the discussion concentrates on the development of the mNOVA-framework and is well referenced, though these references should have been discussed considerably earlier – when the critical assessment of the mNOVA-Framework was initially mentioned. Otherwise, the discussion concentrates on the limitations of the selected methodological approach, which was sparsely referenced. From the discussion, the reader is given the impression that the mNOVA framework's development and its application to the FFQ were the centerpieces of this manuscript. However, this was not matched by the rest of the manuscript.

Author Response

(The authors gave the same response as above.)

Reviewer 3 Report

Comments and Suggestions for Authors

Dear Authors,

The study protocol aims to validate a food frequency questionnaire (FFQ) to assess the consumption of ultra-processed foods (UPF) in the Italian population. The results of the study will provide a valid tool for analyzing dietary patterns and contribute to a better understanding of the impact of UPF on public health.

The study is interesting and current, but I have the following comments:

1. It is necessary to describe how you will avoid false responses in the questionnaire and how you will verify the accuracy of the data?

2. Provide a more detailed description of the study's outcomes and how they will be presented to the public?

3. Do you plan to publish the results in a scientific journal? If so, it should be mentioned in the text.

Author Response

(The authors gave the same response as above.)

Round 2

Reviewer 2 Report

Comments and Suggestions for Authors

Thank you for the constructive engagement with the feedback. I have no further comments.